# Quality Assessment of YUNYAO GNSS-RO Data in the Neutral Atmosphere

Xiaoze Xu[1], Wei Han[2], Jincheng Wang[2], Zhiqiu Gao[3, 1], Fenghui Li[4], Yan Cheng[4], Naifeng Fu[4, 5]

[1]Nanjing University of Information Science and Technology, Nanjing 210044, China
[2]CMA Earth System Modeling and Prediction Centre (CEMC), China Meteorological Administration, Beijing 100081, China
[3]State Key Laboratory of Atmospheric Environment and Extreme Meteorology, Chinese Academy of Sciences, Beijing, 100029, China.
[4]Tianjin Yunyao Aerospace Technology Company, Ltd., Tianjin 300350, China
[5]School of Marine Science and Technology, Tianjin University, Tianjin 300350, China

*Correspondence to*: Wei. Han (hanwei@cma.gov.cn)

**Abstract.** GNSS (Global Navigation Satellite System) Radio Occultation (RO) data are an important component of numerical weather prediction (NWP) systems. To incorporate more GNSS-RO data into NWP systems, commercial RO data has become an excellent option. Tianjin Yunyao Aerospace Technology Co., Ltd. (YUNYAO) plans to launch a meteorological constellation of 90 satellites equipped with GNSS-RO instruments, which will significantly increase the amount of GNSS-RO
data in NWP systems. This study evaluates the quality of neutral atmospheric refractivity and bending angle profiles from YUNYAO satellites Y003 to Y010 during the period from May 1 to July 31, 2023. Compared with the refractivity and bending angle calculated from ERA5, the absolute value of the mean bias (MB) for YUNYAO refractivity and bending angle data are less than 1.54% and 4.51%, respectively, in the height range of 0 to 40 km, and close to 0 between 4 and 40 km. The standard deviations (SD) are below 3.35% and 11.06%, respectively, with variations in SDs among different GNSS satellites, especially
in the lower troposphere and the stratosphere. Second, the refractivity error SD of YUNYAO RO data is estimated using the "three-cornered hat" (3CH) method and multiple data sets. In the pressure range of 1000–10 hPa, the refractivity error SD of YUNYAO RO data is below 2.53%, and the differences in refractivity error SD among different GNSS satellites do not exceed 0.52%. Finally, compared to COSMIC-2 and Metop-C RO data, YUNYAO RO data exhibit consistent refractivity error SD and are smaller within 300–50 hPa.

**1 Introduction**

Radio occultation (RO) of the Global Navigation Satellite System (GNSS) satellites, as observed from low Earth orbiting (LEO) satellites, is used for remote sensing of the Earth's neutral atmosphere and ionosphere (Anthes et al., 2008). When a radio signal from a transmitter on a GNSS satellite passes through the limb of the atmosphere, the timing and direction of the signal received by LEO satellites are different from those of a straight-line path through a vacuum because the signal path is
bent by the vertical gradient of atmospheric refractivity (Rocken et al., 1997). The profile of refractivity can be derived from the profile of bending angle, analytically in the limit of a spherically symmetric atmosphere, using the Abel transform

(Kursinski et al., 1997). The atmospheric refractivity is a function of atmospheric temperature, pressure and water vapor pressure (Kursinski et al., 1997).

GNSS-RO observations can provide temperature information for the stratosphere and upper troposphere and humidity information for the lower troposphere (Eyre et al., 2022). Additionally, they are characterized by high vertical resolution, high accuracy, all-weather capability, and global coverage (Rocken et al., 1997; Schreiner et al., 2020; Sun et al., 2018; Ware et al., 1996), and they exhibit minimal bias between 5 and 30 km (Wickert et al., 2005). Therefore, GNSS-RO data products (i.e., bending angle, refractivity, temperature, water vapor, and pressure) have been widely used in numerical weather prediction (NWP) centers and have shown a significant positive impact on regional and global NWP forecasts (Anthes et al., 2024; Aparicio and Deblonde, 2008; Cucurull and Derber, 2008; E. Bowler, 2020; Harnisch et al., 2013; Healy and Thépaut, 2006; Huang et al., 2010; Le Marshall et al., 2010; Liu and Xue, 2014; Miller et al., 2023; Poli et al., 2008; Ruston and Healy, 2021; Sun et al., 2018), particularly in the upper troposphere and lower stratosphere, and especially in the Southern Hemisphere regions (Cucurull and Derber, 2008; Eyre et al., 2022; Rennie, 2010; Sun et al., 2018).

GNSS-RO ranks among the top contributors in global NWP systems (Cardinali and Healy, 2014; Eyre et al., 2022) and plays an important role as "anchor observations" in the calibration of the radiance bias corrections (Aparicio and Laroche, 2015). Harnisch et al. (2013) found that even with 128,000 RO profiles per day for assimilation, increasing the number of RO profiles is still expected to provide additional benefits to the forecast. As of 2020, the missions providing GNSS-RO data to NWP centers have been summarized in Eyre et al. (2022). Currently, the Global Forecast System (CMA-GFS), developed by the China Meteorological Administration, incorporates approximately 20,000 GNSS-RO profiles per day. As shown in Table 1, this includes data summarized by Eyre et al. (2022) as well as data from FengYun-3E and commercial GNSS-RO data from Spire (Ho et al., 2023). Commercial GNSS-RO data from Spire constitutes 22% of the total number of RO profiles in CMA-GFS, highlighting the importance of commercial RO data in global NWP systems.

**Table 1. The number of RO profiles applied in CMA-GFS on May 25, 2023.**

| Satellite | Metop-B | Metop-C | TerraSAR-X | TanDEM-X | PAZ |
|---|---|---|---|---|---|
| Number | 551 | 520 | 141 | 85 | 137 |
| Satellite | Sentinel-6a | FY-3C | FY-3D | FY-3E | COSMIC-2 |
| Number | 619 | 440 | 607 | 882 | 6272 |
| Satellite | GRACE-C | GRACE-D | Spire | Others | |
| Number | 117 | 142 | 4461 | 5327 | |

Tianjin Yunyao Aerospace Technology Co., Ltd. (YUNYAO) plans to launch an 90-satellite meteorological constellation equipped with GNSS-RO instruments (Table 2, Fu and Li, 2021). The YUNYAO GNSS-RO payload, as a multi-GNSS receiver, can simultaneously receive radio signals from the U.S. Global Positioning System (GPS), the Chinese BeiDou Navigation Satellite System (BDS), the Russian Global Navigation Satellite System (GLONASS), and the Galileo (GAL). YUNYAO RO will significantly increase the number of GNSS-RO observations available to NWP centers and is expected to further improve the accuracy of NWP forecasts.

**Table 2. The launch schedule of YUNYAO satellites.**

| Serial Number | Satellite Name | Launch Date | Orbit Altitude (km) | Orbit Inclination (°) |
|---|---|---|---|---|
| 31–48 | Y013, Y023, Y024, Y027–Y029, Y037–Y048 | Q1 2025 (first quarter of 2025) | 535 | 94.5 |
| 49–60 | Y049–Y060 | Q2 2025 (second quarter of 2025) | 535 | 94.5 |
| 61–90 | Y061–Y90 | Q3–Q4 2025 (third and fourth quarters of 2025) | 500/600/800, 535 | 50, 94.5 |

In this study, we conducted a quality assessment of the neutral atmospheric RO profiles of YUNYAO satellites 003 to 010 from May 1 to July 31, 2023. First, we compared the YUNYAO refractivity and bending angle data with the fifth-generation European Centre for Medium-Range Weather Forecasts (ERA5) reanalysis data. Then, using the "three-cornered hat" method (3CH; Anthes & Rieckh, 2018), we estimated the standard deviation (SD) of the errors in the YUNYAO RO refractivity data. Finally, we compared the assessment results of YUNYAO refractivity data with those of UCAR's COSMIC-2 (hereafter referred to as C2) and Metop-C (hereafter referred to as MTPC).

## 2 Data and Method

### 2.1 Data

### 2.1.1 YUNYAO RO Data

Since the first time the YUNYAO Data Processing Center provided the GNSS-RO profile products for the Radio Occultation Modeling Experiment (ROMEX), their data processing methodology has been updated in three key aspects. First, the deviation observed between 20 and 40 km, distinct from other GNSS-RO missions, was resolved by adjusting the smoothing window width for the exceed phase-to-Doppler inversion to optimize its adaptability to YUNYAO's high-sampling-rate data (100 Hz). Second, to address the sudden increase in SD below 12 km, YUNYAO investigated the open-/close-loop transition algorithms employed in other GNSS-RO missions. Their retrieval chain automatically identified the L1 open-/closed-loop splicing points through the L2 lock marks and used a sigmoid function as a weight to ensure a smooth transition from L1 closed-loop observations to open-loop observations. Third, for altitudes below 5 km, YUNYAO redesigned the L1 data truncation strategy to use the complete L1 open-loop observations as much as possible and process to obtain continuous exceed phases. In the geometric optics retrieval process, the Doppler retrieval truncation strategy is implemented by identifying cases where the difference between the Doppler shift obtained from L1 and that from the empirical atmospheric model exceeds a specified threshold. In the wave optics retrieval process, the effective bending angle sequence is obtained by restoring the signal amplitude from the exceed phase at each height in the full spectrum inversion of the bending angle retrieval.

YUNYAO RO data used in this study are from May 1 to July 31, 2023. During the quality evaluation period, a total of eight LEO satellites provided RO observations (hereafter referred to as Y003, Y004, Y005, Y006, Y007, Y008, Y009, and Y010).

These LEO satellites are all in high-inclination-angle orbits, allowing their observations to cover the entire globe. The designed parameters of YUNYAO GNSS receiver are summarized in Table 3. YUNYAO GNSS receivers are compact in size, with their weight being approximately one-nineteenth that of the FengYun-3C/GNOS (Sun et al., 2018).

**Table 3. Characteristics of the YUNYAO GNSS receiver.**

| Parameters | Content |
|---|---|
| GNSS signals | BDS B1 |
| | BDS B3 |
| | GPS L1 |
| | GPS L2 |
| | GLONASS G1 |
| | GLONASS G2 |
| | GAL E1 |
| | GAL E5b |
| Size | $115 \times 155 \times 60$ mm$^3$ |
| Weight | 0.8 kg |
| Power | $\leq 12$ W |

Fig. 1 shows the time series of the number of RO profiles for these eight satellites. It is important to note that this study only

90 evaluates the RO profiles from GPS, BDS, and GLONASS. The number of RO profiles generated daily by the eight satellites is almost the same, but there are differences in the number of profiles from different GNSS satellites. The number of BDS RO profiles is the highest, while the number of GLONASS RO profiles is the lowest. Specifically, each satellite produces an average of approximately 1500 RO profiles per day, with around 500 from GPS, 350 from GLONASS, and 650 from BDS. Taking Y008 as an example, Fig. 2b, Fig. 2d, and Fig. 2f show the horizontal distribution of the number of GPS, GLONASS,

and BDS RO profiles, respectively. As shown in the figure, RO profiles cover the entire globe. Notably, the data transmission of the YUNYAO satellite primarily depends on ground stations situated within China. During data transmission, the satellite is required to perform specific onboard operations, leading to a reduction in occultation observations before entering Chinese airspace, particularly over the Indonesia region. Furthermore, due to GNSS signal interference in Europe and the Middle East, particularly with L2 signals, the number of occultation observations decreases. In the Central America region, the reduced

number of BDS occultation observations correlates with the number of BDS satellites.

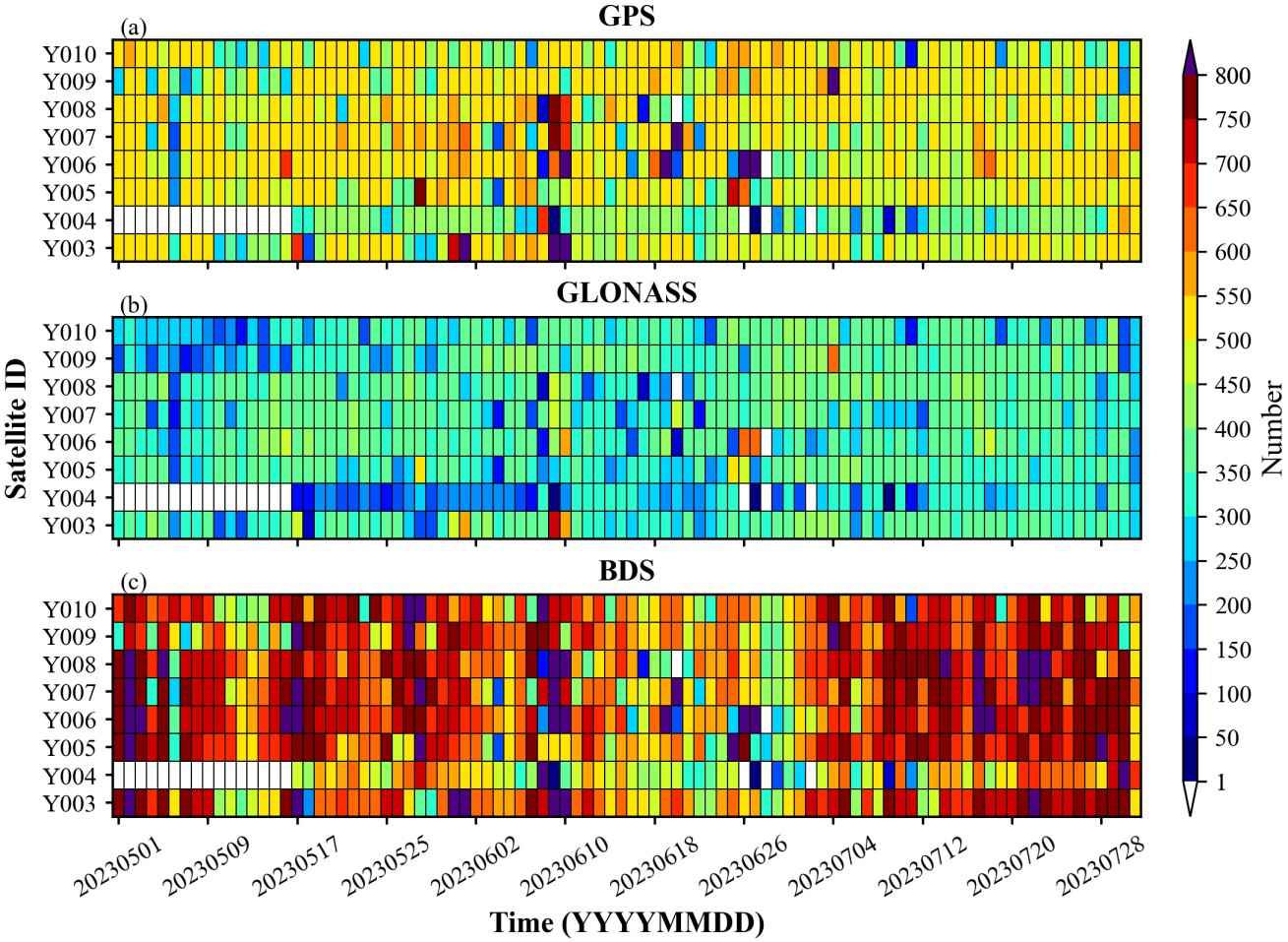

**Figure 1: The time series of the number of RO profiles for Y003-Y010 (a, b, c represent GPS, GLONASS, and BDS occultations, respectively).**

Fig. 2a, Fig. 2c, and Fig. 2e show the penetration depths of Y008 GPS, GLONASS, and BDS occultation profiles, respectively.

Y008 satellite was selected randomly, and the performance of the other YUNYAO satellites is consistent with Y008. The penetration depth of RO is an important indicator of its detection performance. Due to the influence of moisture, various errors such as multipath propagation errors, receiver tracking errors, and super-refraction errors are introduced in the lower troposphere. Therefore, the detection capability of signal in the lower troposphere is limited, and the proportion of profiles that can penetrate the complex atmosphere to reach near the ground is reduced. As shown in Fig. 2, the RO profiles from BDS and

GPS have consistent penetration depth, while the penetration depth of RO profiles from GLONASS is not as deep as those from BDS and GPS. In mid-to-high latitude regions, the penetration depth of BDS and GPS RO profiles is mostly below 1 km, while in low-latitude regions, it is around 2 km. This may be related to the higher humidity levels in low-latitude regions. For GLONASS RO profiles, the penetration depth is around 2 km in high-latitude regions and around 4 km in low-latitude regions.

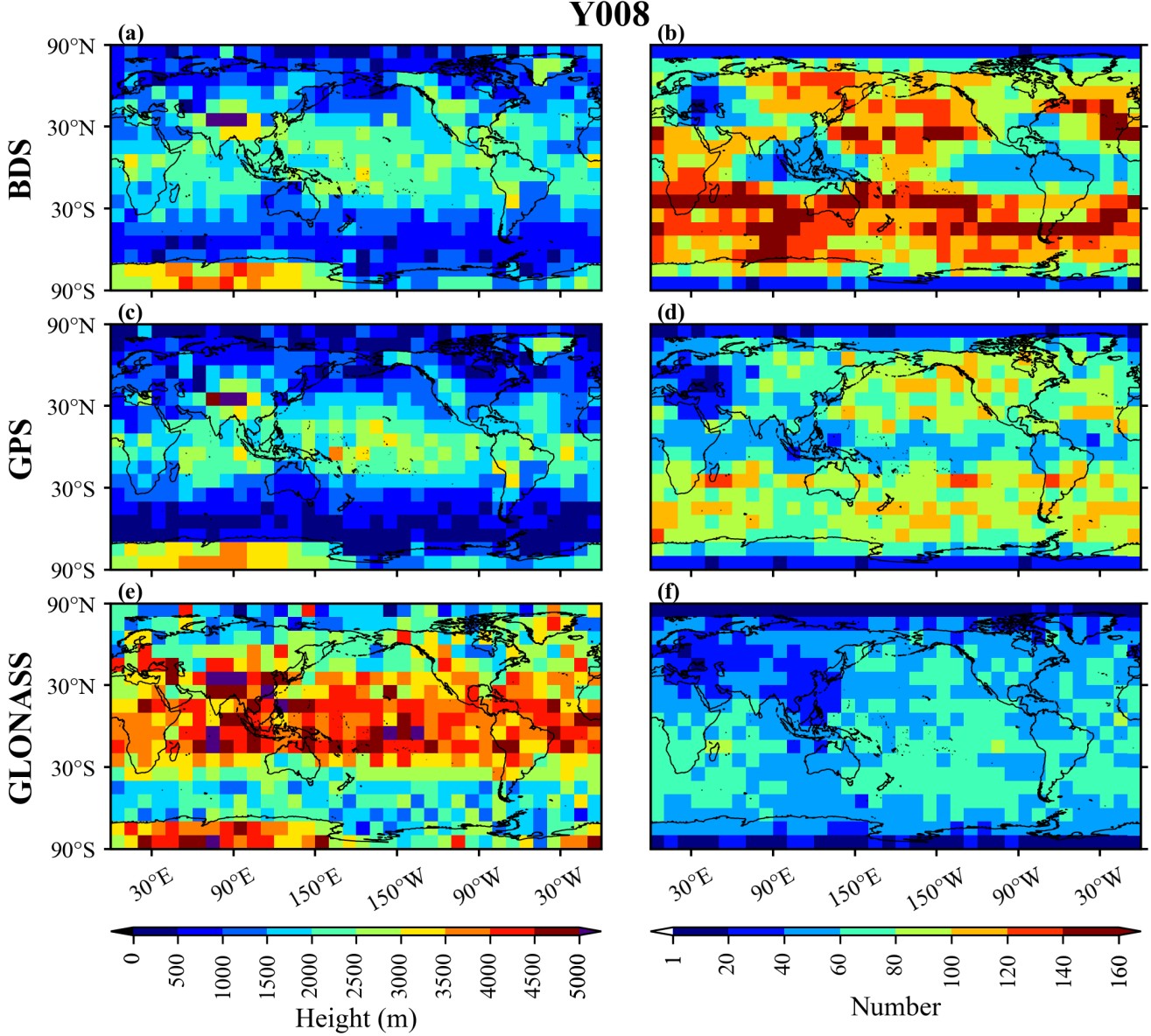

**Figure 2: The spatial distribution of the average penetration depths of Y008 RO profiles (a, c, and e represent GPS, GLONASS, and BDS, respectively) and the spatial distribution of the number of RO profiles (b, d, and f represent GPS, GLONASS, and BDS, respectively). RO profiles are grouped by their latitude and longitude positions into 10°×10° grid boxes for statistical analysis.**

 **2.1.2 COSMIC-2 and Metop-C RO Data**

This study compares the assessment results of C2 and MTPC RO profiles with the assessment results of YUNYAO RO profiles to analyze the quality of YUNYAO RO profiles. C2 and Metop/GRAO RO data have been extensively evaluated and have shown positive impacts on numerical weather prediction (Anlauf et al., 2011; Cucurull, 2023; Schreiner et al., 2011, 2020). C2 and MTPC RO data are obtained from the COSMIC Data Analysis and Archive Center (CDAAC, www.cosmic.ucar.edu). Similar to the period of YUNYAO RO data, C2 and MTPC RO data from May 1, 2023, to July 31, 2023, were used.

**2.1.3 ERA5 Data**

ERA5 is the fifth generation of the ECMWF reanalysis dataset, available in the Climate Data Store (https://cds.climate.copernicus.eu/). The ERA5 data provide hourly atmospheric parameters with a horizontal grid spacing of $0.25° × 0.25°$ and 37 pressure levels ranging from 1,000 to 1 hPa. ERA5 data are widely regarded as the most comprehensive and accurate reanalysis archive. Therefore, we first compared the YUNYAO RO profiles with the ERA5 reanalysis data. ERA5 data were also used in the 3CH method. This study used ERA5 data only from 00, 06, 12, and 18 UTC.

**2.1.4 FNL Data**

FNL is the final operational global analysis data from the Global Forecasting System of the National Centers for Environmental Prediction (NCEP). The FNL is produced using the same model that NCEP uses in the Global Forecast System (GFS), but the FNLs are prepared about an hour or so after the GFS is initialized to allow for the use of more observational data. The FNL analysis data are published every 6 hours on a $1° × 1°$ global latitude-longitude grid and include 26 mandatory (and other pressure) levels from 1000 hPa to 10 hPa (https://rda.ucar.edu/datasets/ds083.2/).

**2.1.5 Radiosonde data**

Balloon sounding is a well-established method for obtaining atmospheric temperature and humidity profiles, and its data quality has been extensively studied and frequently used as a reference standard for evaluating other soundings (Corner et al., 1999; Lanzante, 1996; Mapes et al., 2003; Miloshevich et al., 2001). Therefore, global radiosonde (RS) observations are used in 3CH method. RS data are usually available twice a day (mostly at 00 and 12 UTC). This study employed only sixteen mandatory pressure levels: 1000, 925, 850, 700, 500, 400, 300, 250, 200, 150, 100, 70, 50, 30, 20, and 10 hPa.

**2.2 Method**

**2.2.1 Observation operator**

The raw observation of RO is the time delay of the radio signal. Through a series of inversions, additional phase, bending angle, refractivity, and atmospheric elements such as temperature, pressure, and humidity can be obtained. The aim of this study is to assess the quality of YUNYAO RO refractivity and bending angle profiles. To facilitate the comparison between

RO data and ERA5/FNL/RS data, it is necessary to transform ERA5/FNL/RS variables into RO variables using an observation operator. This study employed the Smith & Weintraub (1953) equation to calculate refractivity ($N$):

$$N = 77.6 \times \frac{P}{T} + 3.73 \times 10^5 \times \frac{e}{T^2}, \tag{1}$$

where $P$ is pressure, $T$ is temperature, $e$ is water vapor pressure. This expression is accurate to 0.5% in $N$ for frequencies up to 30,000 MHz and under typical ranges of temperature, pressure, and humidity (A more detailed description can be found in Smith & Weintraub (1953)).

The one-dimensional forward model in the Radio Occultation Processing Package (ROPP) was used to transform the ERA5 variables into bending angles. The forward modelling can be simplified by assuming spherical symmetry, so that the impact parameter remains constant along the ray path. In this case, it can be shown that the bending angle is given by the following integral:

$$\alpha(a) = -2a \int_a^\infty \frac{1}{\sqrt{x^2-a^2}} \frac{d\ln(n)}{dx} dx, \tag{2}$$

where $\alpha$ is bending angle, $a$ is impact parameter, $n$ is refractive index and $x = nr$ (r is the geocentric radial distance).

## 2.2.2 Comparison with ERA5

The evaluation method for this section involves analyzing the mean bias (MB) and SD between the RO refractivity and bending angle data and the those calculated from ERA5. Given that the magnitude of refractivity and bending angle data decreases exponentially with height, the MB is normalized using the refractivity and bending angle calculated from ERA5, specifically analyzing $\frac{N^o-N^b}{N^b}$ and $\frac{\alpha^o-\alpha^b}{\alpha^b}$, where $N^o$ represents the observed refractivity, $N^b$ represents the refractivity calculated from ERA5, $\alpha^o$ represents the observed bending angle and $\alpha^b$ represents the bending angle calculated from ERA5.

Given the different temporal and spatial resolutions of RO data and ERA5 reanalysis data, it is necessary to perform temporal and spatial matching to enable comparison. For temporal matching, ERA5 data are linearly interpolated to the RO observation times. For spatial matching, ERA5 data are interpolated to the positions of the RO observations using bilinear interpolation in the horizontal direction. Then, the refractivity at 37 levels of ERA5 is calculated using Equation (1), and the bending angles are computed at heights of every 400 geopotential meters from 0 to 60,000 geopotential meters using the one-dimensional ROPP model (60,000 meters is the default setting, the calculation is actually performed only up to the highest altitude of the ERA5 profile). Since the vertical resolution of RO is significantly higher than the vertical resolution of the refractivity and bending angle profiles calculated from ERA5 data, we first perform a linear interpolation of the RO observations to the heights of the refractivity and bending angle profiles derived from ERA5 data and calculate the biases. Then, the calculated biases are linearly interpolated to heights from 0 to 40 km at 400-meter intervals. The use of linear interpolation assumes that the bias varies linearly with height, although the refractivity and bending angle do not change linearly with height.

Due to errors in electromagnetic wave signals and other reasons, there may be erroneous RO data or outlier observations that are far from the simulated values. Therefore, the bi-weight method was used to eliminate outliers (Lanzante, 1996). Similar to

Zou & Zeng (2006), this study used c=7.5 and a Z-score threshold of 4. The data were divided into three latitudinal bands for quality control: low latitude (30°S to 30°N), midlatitude (30°N to 60°N and 30°S to 60°S), and high latitude (60°N to 90°N and 60°S to 90°S). Observations that passed quality control were used to calculate the biases. Additionally, for refractivity biases, if a single RO refractivity profile contains observations with biases exceeding 50% within 10 to 30 km, the entire profile is discarded. If more than 50% of the observations in a single refractivity RO profile have biases exceeding 20%, the entire profile is also discarded. Finally, observations with RO biases exceeding 10% were eliminated. For bending angle biases, the same quality control method was applied, although RO observations with bending angle biases exceeding 10% were retained. For all quality-controlled samples, the MB and SD are calculated using the following formulas:

$$X_i = \frac{N_i^o - N_i^b}{N_i^b} \ or \ \frac{\alpha_i^o - \alpha_i^b}{\alpha_i^b}, \tag{3}$$

$$MB = \frac{\sum_{i=1}^{n} X_i}{n}, \tag{4}$$

$$SD = \sqrt{\frac{\sum_{i=1}^{n}(X_i - MB)^2}{n}}, \tag{5}$$

where $X_i$ represents the refractivity bias or bending angle bias of a single sample and $n$ represents the total number of samples.

### 2.2.3 3CH Method

In this study, the 3CH method (Anthes & Rieckh, 2018) is used to estimate the random error SD (uncertainty) of RO refractivity observations. Anthes & Rieckh (2018) provided a detailed description of the 3CH method. Similar to Anthes & Rieckh (2018), three data sets (ERA5, FNL, and RS) are used to estimate the errors of RO observations. The 3CH equations include bias correction terms, we remove the mean biases from each data set (O'Carroll et al., 2008).

ERA5 and FNL data are interpolated to the locations and times of RO data, and only the 16 pressure levels corresponding to the mandatory levels of RS are used. Then, the RO data are vertically interpolated to the heights of ERA5. RO data within 3 hours and 300 km of the RS locations are matched to the RS data. Considering the spatial and temporal differences between RS and the matched RO observations, we applied a spatial-temporal sampling correction (Gilpin et al., 2018). The specific approach is as follows: first, the ERA5 data are interpolated to the times and locations of both RO and RS data to calculate the refractivity. Then, the difference in refractivity between these two spatiotemporal positions is computed to represent the bias introduced by spatiotemporal differences. Finally, this bias is subtracted from the refractivity calculated from the RS data. After performing the above steps, the error SDs for YUNYAO, C2, and MTPC are estimated at the 16 mandatory pressure levels of RS.

It is worth noting that this study used three data sets to estimate the RO errors, which allows for the production of three independent 3CH estimates of the error SD. The results section mainly discusses the mean of these three error SDs.

# 3 Results

## 3.1 Comparison with ERA5

As described in Section 2.1.1, a total of eight satellites provided RO observations during the study period, and each satellite was capable of receiving radio signals from BDS, GPS, and GLONASS simultaneously. The RO observations from different LEO satellites and different GNSS satellites were evaluated separately. Fig. 3 shows the comparison of YUNYAO RO refractivity profiles with ERA5. Each subplot represents a YUNYAO satellite, with black, red, and blue colors representing BDS, GPS, and GLONASS, respectively. For each YUNYAO satellite, the numbers of BDS, GPS, and GLONASS occultation

profiles used for comparison is approximately 31,000, 41,000, and 59,000, respectively, with relatively fewer profiles for the Y004 satellite.

The RO refractivity data from each YUNYAO satellite exhibits similar bias characteristics. Taking the Y008 satellite as an example, the absolute values of MB for BDS, GPS and GLONASS are less than 1.46%, 1.31%, and 0.86%, respectively. Below 4 km, there is a slight negative bias, and GLONASS exhibits the smallest bias. RO biases in the lower troposphere are

220 caused by various factors, such as super-refraction (Ao, 2007; Ao et al., 2003; Sokolovskiy, 2003; Xie et al., 2006), tracking depth and noise (Sokolovskiy, 2003; Sokolovskiy et al., 2010), and fluctuations of refractivity (Gorbunov et al., 2015; Gorbunov and Kirchengast, 2018). The MB is minimal between 4 and 40 km. Specifically, the average values of MB for BDS, GPS, and GLONASS between 4 and 40 km are 0.022%, 0.036%, and –0.008%, respectively, while the average values of MB between 10 and 30 km are 0.020%, 0.026%, and –0.019%, respectively.

The SDs of BDS, GPS, and GLONASS are less than 2.58%, 2.60%, and 3.34%, respectively, with larger SDs occurring in the lower troposphere and upper stratosphere. The increase in uncertainty in the upper stratosphere is related to the reduction of the neutral atmospheric signal below the phase noise level (Sokolovskiy et al., 2010). The SDs of BDS, GPS, and GLONASS between 10 and 30 km are less than 1.09%, 1.17%, and 1.24%, respectively. Notably, there are differences in the SD of BDS, GPS, and GLONASS, especially in the lower troposphere and upper stratosphere. Below 4 km, the SD of GLONASS is the

smallest. Between 4 km and 30 km, the SDs of BDS, GPS, and GLONASS are relatively consistent. Above 30 km, the SDs clearly show the pattern BDS < GPS < GLONASS.

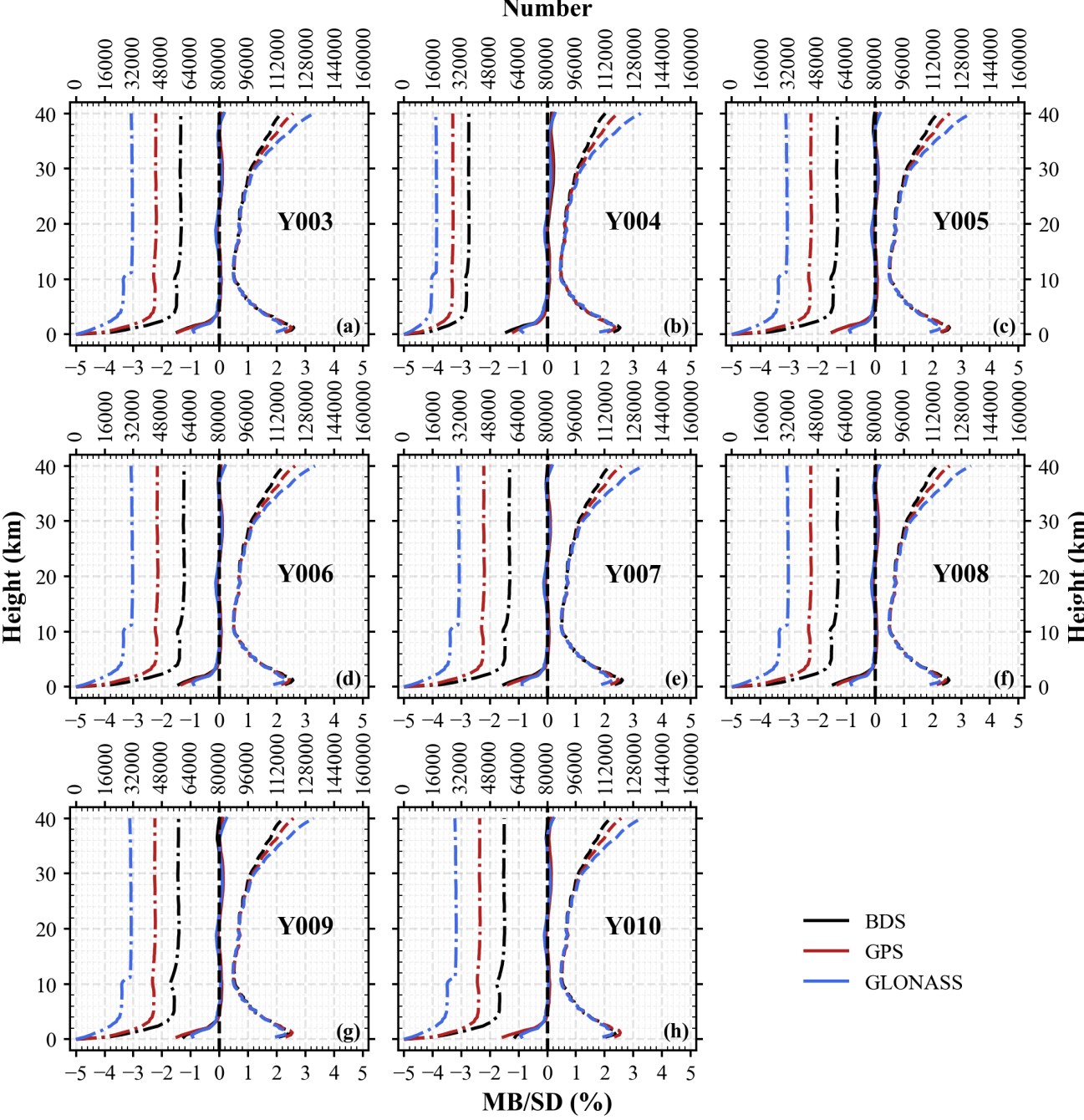

**Figure 3: Comparison of Y003-Y010 RO refractivity profiles with ERA5. The solid, dashed, and dash-doted lines** represent MB, SD, and observation number, respectively. Black, red, and blue colors represent BDS, GPS, and GLONASS, respectively. The horizontal axis below each subplot is used for MB and SD, while the horizontal axis above is used for observation number.

To further discuss the horizontal distribution of data quality, Fig. 4 shows the variation of MB and SD with latitude between the Y008 BDS refractivity profiles and ERA5. As shown in Fig. 4, the Y008 BDS refractivity profiles exhibit a negative bias in the lower troposphere, with larger biases in low-latitude regions. Above 4 km, a cold bias still exists in low-latitude regions, and the bias gradually increases with altitude. In mid-to-high latitude regions, there is mainly a positive bias, which also gradually increases with altitude. The areas with larger SD are mainly distributed in two places: the lower troposphere in low-latitude regions and the stratosphere in the southern hemisphere. The negative bias and larger SD in the lower troposphere in low-latitude regions are related to higher water vapor content, while the larger SD in the southern hemisphere stratosphere may originate from model biases (Cucurull et al., 2007).

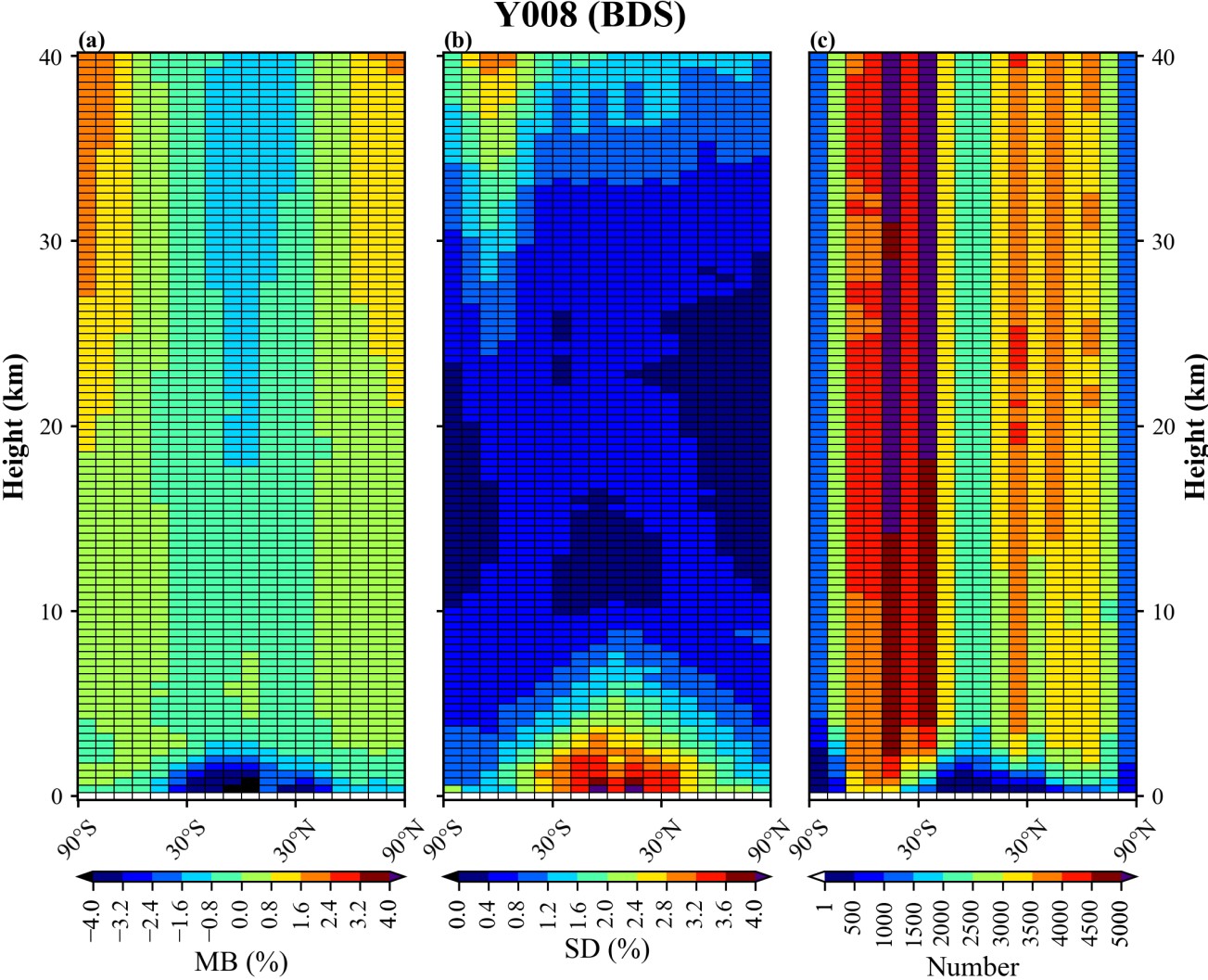

**Figure 4: The latitude variation of the MB (a) and SD (b) of Y008 BDS refractivity profiles compared to ERA5, as well as the latitude variation of observation number (c).**

The bias characteristics of bending angles closely resemble those of refractivity (Fig. 5). Taking the Y008 satellite as an example, the absolute values of MB for BDS, GPS and GLONASS are less than 4.31%, 3.58%, and 1.31%, respectively. Between 4 and 40 km, the average values of MB for BDS, GPS, and GLONASS are 0.15%, 0.15%, and 0.09%, respectively. Between 10 and 30 km, the average values of MB are 0.12%, 0.13%, and 0.06%, respectively. Between 0 and 40 km, the SDs of BDS, GPS, and GLONASS are less than 10.87%, 10.00%, and 9.50%, respectively. Between 10 and 30 km, the SDs of BDS, GPS, and GLONASS are less than 2.09%, 2.11%, and 2.23%, respectively. Notably, the wavy structures in MB and SD are related to the sparse vertical layering of ERA5. The MB and SD of refractivity do not exhibit such phenomena because the interpolation method avoids directly interpolating the refractivity calculated from ERA5 to high resolution (see Section 3.1). In contrast, the bending angle is directly calculated at high resolution and is an integral quantity. The latitudinal distribution of bending angle biases aligns with the distribution of refractivity (Figure S1).

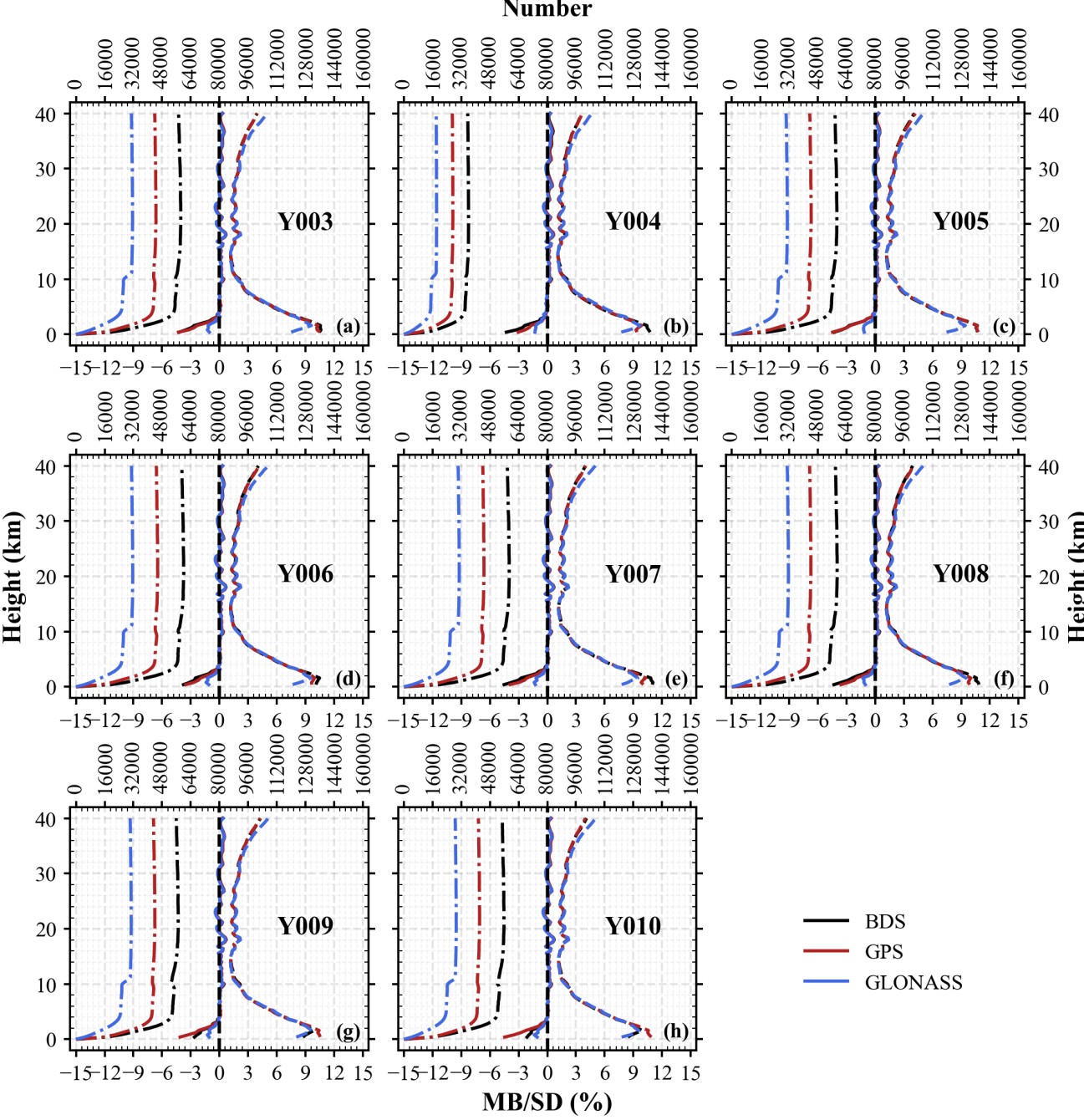

**Figure 5: Comparison of Y003-Y010 RO bending angle profiles with ERA5. The solid, dashed, and dash-doted lines represent MB, SD, and observation number, respectively. Black, red, and blue colors represent BDS, GPS, and GLONASS, respectively. The horizontal axis below each subplot is used for MB and SD, while the horizontal axis above is used for observation number.**

## 3.2 3CH Results

ERA5, FNL, RS, and YUNYAO RO data were used in the 3CH method. Four data sets produce three independent 3CH estimates of the error SD. Fig. 6 shows the 3CH results for the four data sets. LEO satellites and GNSS satellites are not distinguished here. In Figure 6a, a spatial-temporal sampling correction was implemented for the RS data, whereas in Figure 6b, the RS data did not undergo this correction. As shown in Fig. 6a, ERA5 demonstrated the smallest error SD, followed by FNL and RS, while RO exhibited the largest error SD. Compared to the results of Schreiner et al. (2020), the error SD of YUNYAO RO is generally consistent with that of C2 (a detailed comparison is provided in the next section). Unlike previous studies (Rieckh et al., 2021; Schreiner et al. 2020), the RS error SD in this study is smaller than the RO error SD, which is due to the application of the spatial-temporal sampling correction. Fig. 6b shows the results in the absence of the spatial-temporal sampling correction for the RS data. As illustrated in the figure, the RS error SD increases significantly. The error SD of YUNYAO RO remains largely unchanged and is smaller than the RS error SD below 10 km. Above 10 km, the error SDs of YUNYAO RO and RS are comparable, which may be due to different quality control applied to the RS data. Notably, the error SD of ERA5 in Fig. 6b is larger than in Fig. 6a. This is because the spatial-temporal sampling correction was implemented using ERA5 data, leading to a notably smaller error SD for ERA5 when estimated with RS data after applying the correction (Fig. S2).

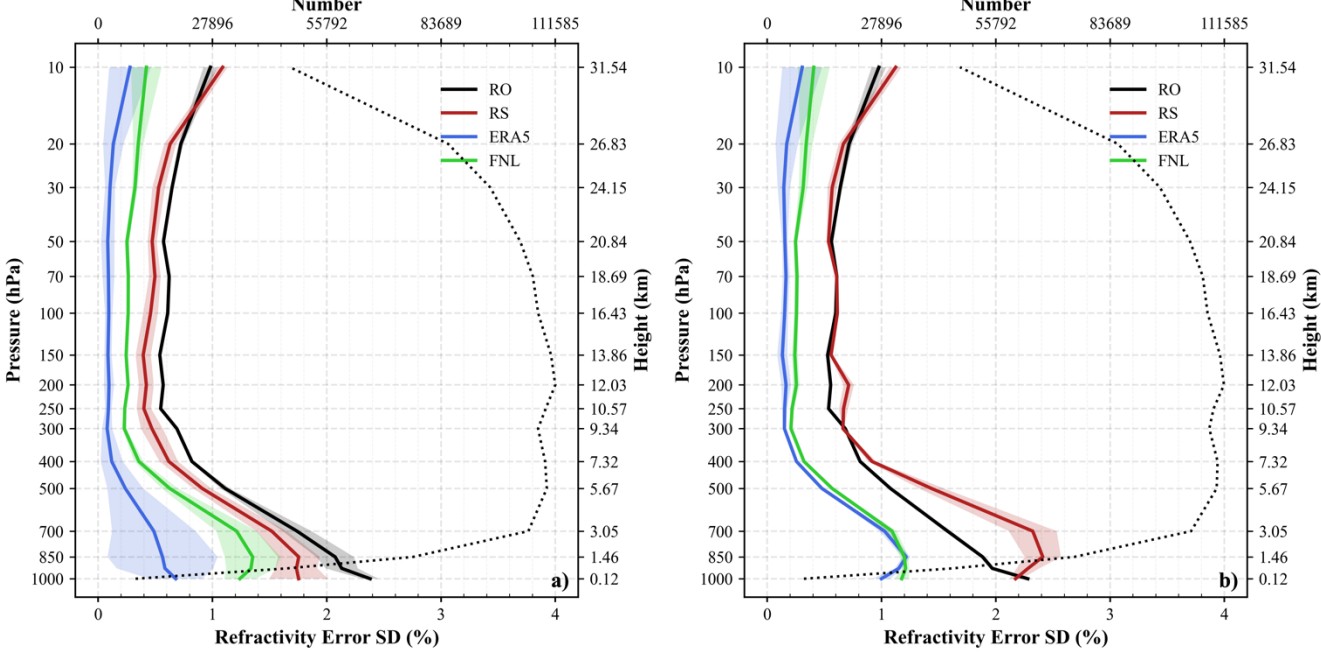

**Figure 6: 3CH estimates of refractivity error SD of YUNYAO RO (black), ERA5 (blue), FNL (green), and RS (red). (a) spatial-temporal sampling correction applied, (b) without spatial-temporal sampling correction. The solid lines**

represent the mean of the three SD estimates for each data set, and the shading indicates the SD of the three error SD estimates. The dotted line shows the number of matched data. The right vertical axis represents the average height of all samples at each pressure level.

The results for different LEO satellites and GNSS satellites are shown in Fig. 7. Similar to Fig. 3, the quality of RO data from different LEO satellites is generally consistent, as evidenced by similar refractivity error SD. In the pressure range of 1000–10 hPa (approximately 0–32 km), the refractivity error SDs of Y002–Y010 are all below 2.53%. Among these, it is greater than 1% in the pressure range of 1000–500 hPa (approximately 0–6 km) and less than 1% in the pressure range of 500–20 hPa (approximately 6–27 km).

There are differences in the refractivity error standard deviation (SD) of RO among different GNSS satellites. Below the 700 hPa level, the refractivity error SD for Y003–Y008 follows the pattern GLONASS < GPS < BDS, while for Y009 and Y010, it follows the pattern GLONASS < BDS < GPS. Around the 300 hPa level, the refractivity error SD for Y003–Y010 follows the pattern GLONASS < GPS < BDS, with the differences being more pronounced for Y009 and Y010. Above the 50 hPa level, the refractivity error SD for Y003–Y010 follows the pattern BDS < GPS < GLONASS. It is worth noting that the differences in refractivity error SD among different GNSS satellites are very small, with the maximum difference not exceeding 0.52%.

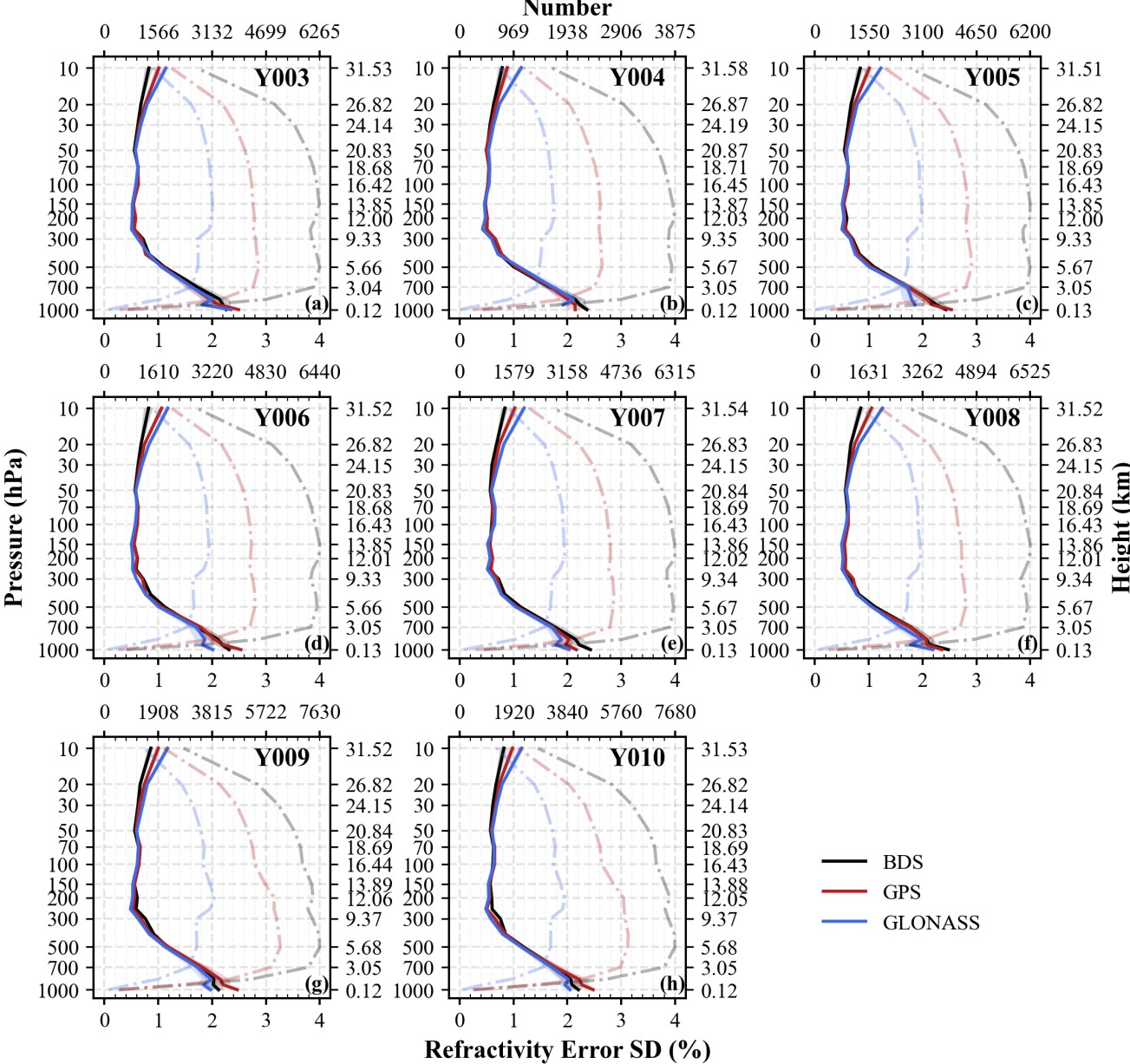

Figure 7: 3CH estimates of refractivity error SD of Y003–Y010 RO. The solid and dash-dotted lines represent refractivity error SD and matched data number. Black, red, and blue colors represent BDS, GPS, and GLONASS, respectively. The horizontal axis below each subplot is used for refractivity error SD, while the horizontal axis above is used for data number. The right vertical axis represents the average height of all samples at each pressure level.

### 3.3 Comparison with Metop-C and COSMIC-2

In this section, we compare the refractivity data of C2 and MTPC with ERA5 and estimate the refractivity error SDs of C2 and MTPC using the 3CH method, and subsequently compare the results of YUNYAO with these results. MTPC observations come from only one LEO satellite. To ensure a fair comparison, we only used one YUNYAO satellite (Y008) and one C2 satellite (the first C2 satellite, hereafter referred to as C2E1). The results of sections 3.1 and 3.2 indicate that the differences among the eight YUNYAO satellites are small, and the study by Schreiner et al., (2020) also indicates that the differences

among the six C2 satellites are small. Additionally, since C2 observations are primarily distributed in the tropics (approximately 45°S to 45°N; Ho et al., 2023), the comparisons between Y008 and C2E1 and between Y008 and MTPC were conducted separately. In the comparison between Y008 and C2E1, only observations between 45°S and 45°N were selected, while in the comparison between Y008 and MTPC, observations covering the entire globe were used.

Fig. 8 shows the comparison of Y008, MTPC and C2E1 RO refractivity profiles with ERA5, with the results of different GNSS

satellites presented separately. As shown in Fig. 8a, Y008 acquires a significantly greater number of RO profiles than MTPC. In comparison to MTPC, Y008 displays a more pronounced negative refractivity bias below 4 km, except for the lesser GLONASS bias below 1 km, while the absolute value of the MB is smaller above 4 km. Regarding SD, Y008 GPS shows smaller values than MTPC GPS in the 10–20 km range. Although the SD of Y008 GPS is greater than that of MTPC GPS above 20 km, the maximum difference does not surpass 1%. As shown in Figure 8b, C2E1 has more profiles in this latitude

range, but Y008 has more profiles globally (not shown). Y008 exhibits a greater negative bias below 5 km, except for GLONASS below 1 km, with the absolute value of MB being smaller above 5 km. Below 25 km, the SD of Y008 aligns with that of C2E1. Above 25 km, the SD of Y008 GPS is larger than that of C2E1 GPS, with a maximum difference of 0.89%. Additionally, the SD of Y008 GLONASS differs from that of C2E1 GLONASS by less than 1%. The SD of Y008 BDS is consistent with that of C2E1 GPS and is smaller than that of C2E1 GLONASS.

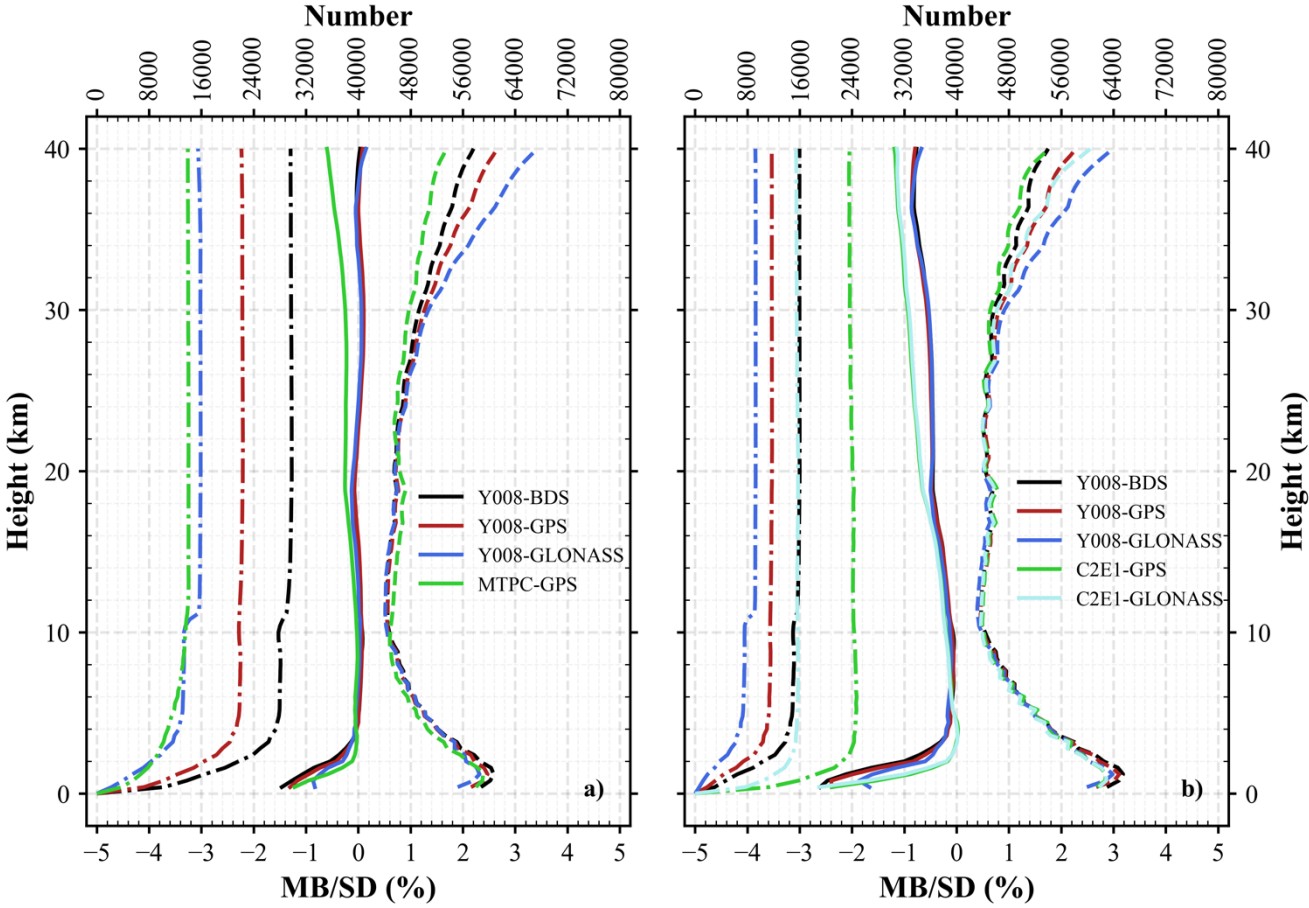

**Figure 8: Comparison of Y008, MTPC and C2E1 RO refractivity profiles with ERA5. The solid, dashed, and dash-doted lines represent MB, SD, and observation number, respectively. The horizontal axis below each subplot is used for MB and SD, while the horizontal axis above is used for observation number. In (a), black, red, blue, and green represent Y008 BDS, Y008 GPS, Y008 GLONASS, and MTPC GPS, respectively. In (b), black, red, blue, green, and**

330 **pale turquoise represent Y008 BDS, Y008 GPS, Y008 GLONASS, C2E1 GPS, and C2E1 GLONASS, respectively.**

Fig. 9 shows the refractivity error SDs of Y008, MTPC, and C2E1, with the error SDs of different GNSS satellites calculated separately. In the pressure range of 1000–300 hPa (approximately 0–9 km), the refractivity error SD of Y008 GLONASS is comparable to that of MTPC GPS, while those of Y008 BDS and GPS are slightly larger. In the pressure range of 300–50 hPa (approximately 9–20 km), the refractivity error SD of Y008 BDS, GPS, and GLONASS is significantly smaller than that of

335 MTPC GPS, consistent with the results presented in Fig. 8a. Notably, in this height range, the number of MTPC samples decreases significantly, mainly due to the fact that some of the matched RS stations lack observations at 150 and 100 hPa (as shown in the red boxes of Figs. S1, S2 and S3). Fewer matched Y008 samples are present at these stations, resulting in no significant reduction in Y008 data within this height range. C2E1 has a greater number of samples matched to these stations,

leading to a more pronounced reduction in data within this height range (as shown in Fig. 9b). In the pressure range of 50–10
340    hPa (approximately 20–32 km), the refractivity error SD of Y008 GPS is larger than that of MTPC GPS, with a maximum
difference of 0.35%. As shown in Fig. 9b, Y008 and C2E1 exhibit comparable error SDs in the pressure range of 1000–300
hPa (approximately 0–10 km). In the pressure range of 300–10 hPa (approximately 10–31 km), Y008 GPS shows smaller
refractivity error SDs than C2E1 GPS, and Y008 GLONASS also exhibits smaller refractivity error SDs compared to C2E1
GLONASS.

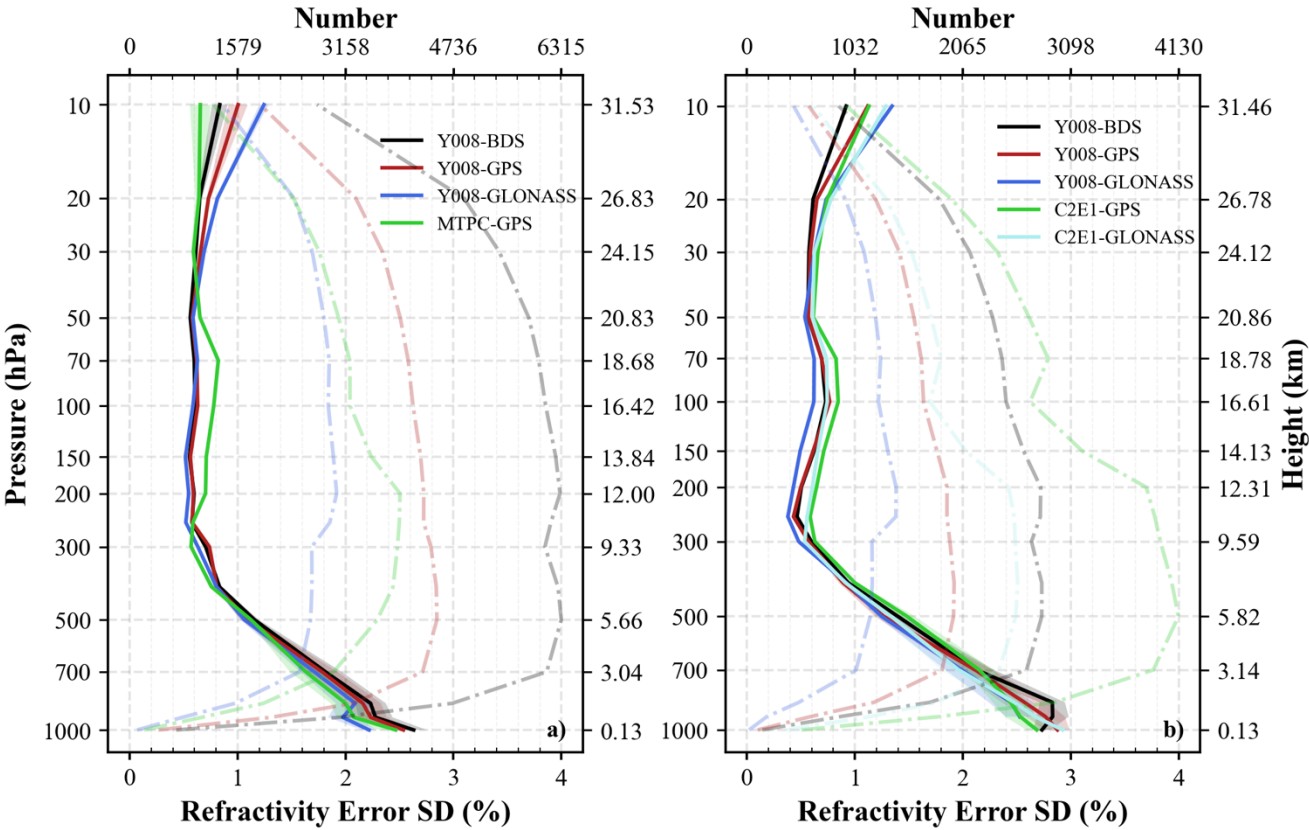

**Figure 9: Comparison of refractivity error SD between Y008 RO and MTPC (a), as well as comparison of refractivity
error SD between Y008 RO and C2E1 (b). The solid and dash-dotted lines represent refractivity error SD and data
number, respectively. The horizontal axis below each subplot is used for refractivity error SD, while the horizontal axis
above is used for data number. The right vertical axis of each subplot represents the average height of all samples at
each pressure level. In (a), black, red, blue, and green represent Y008 BDS, Y008 GPS, Y008 GLONASS, and MTPC
GPS, respectively. In (b), black, red, blue, green, and pale turquoise represent Y008 BDS, Y008 GPS, Y008 GLONASS,
C2E1 GPS, and C2E1 GLONASS, respectively.**

## 4 Summary and conclusions

Tianjin Yunyao Aerospace Technology Co., Ltd. (YUNYAO) plans to establish a meteorological satellite constellation consisting of 90 satellites equipped with GNSS-RO instruments. As of 2024, 30 satellites have been successfully launched. To investigate the usability of YUNYAO RO data, this study evaluated the quality of the YUNYAO RO refractivity and bending angle data. The assessment data includes GNSS-RO data obtained from 8 satellites over the three-month period from May to July 2023.

Compared to the refractivity and bending angle calculated from ERA5, the absolute values of the mean bias (MB) for YUNYAO RO refractivity and bending angle data within the 0–40 km range are less than 1.54% and 4.51%, respectively, and close to 0 between 10 and 30 km. Larger biases are primarily observed in the lower troposphere, a phenomenon that has been extensively discussed in previous studies (Sokolovskiy et al., 2014; Xie et al., 2010). The standard deviation (SD) of refractivity and bending angle data between 0 and 40 km are less than 3.35% and 11.06%, respectively, and is less than 1.24% and 2.27%, respectively, between 10 and 30 km. The increased uncertainty in the lower troposphere is primarily related to the reduction of the neutral atmospheric signal below the noise level in terms of the amplitude, while the increased uncertainty in the upper stratosphere is associated with the reduction of the neutral atmospheric signal below the noise level in terms of the phase (Sokolovskiy et al., 2010). The study also found that the SDs of different GNSS satellites above 30 km show differences, with GLONASS having the largest SD. Latitudinal differences in MB and SD are evident. Larger absolute values of MB are primarily observed in the lower troposphere of low-latitude regions and the upper stratosphere, and this latitudinal distribution of the MB aligns closely with the discussion by Xu and Zou (2020) on bending angles. Larger SDs are mainly distributed in the lower troposphere of low-latitude regions and the stratosphere in the Southern Hemisphere. The abnormally larger SD in the stratosphere of the Southern Hemisphere may be related to the lower skill of ERA5 in simulating refractivity in the Southern Hemisphere at these high altitudes (Gilpin et al., 2018).

This study also used the "three-cornered hat" (3CH) method to estimate the error SD of YUNYAO RO refractivity data. The refractivity error SD of YUNYAO is below 2.53% within the 1000–10 hPa pressure range. The refractivity error SD of YUNYAO is generally consistent with the evaluation results of COSMIC-2 by Schreiner et al. (2020). The refractivity error SD of RS is smaller than that of YUNYAO, which differs from previous studies (Rieckh et al., 2021; Schreiner et al., 2020). This difference is primarily due to the application of spatial-temporal sampling correction to the RS data, and the inconsistencies observed at other altitudes may result from variations in quality control processes. Similarly, differences in refractivity error SD are observed among different GNSS satellites, but the differences are less than 0.52%. In comparison with COSMIC-2 and Metop-C RO data, YUNYAO RO data show a larger negative bias in the lower troposphere but smaller biases at other altitudes. The refractivity error SD of YUNYAO RO data is consistent with that of COSMIC-2 and Metop-C RO data within the 1000–10 hPa pressure range, and is smaller within the 300–50 hPa pressure range.

YUNYAO commercial RO small satellites are characterized by their small size and low cost, while maintaining high detection accuracy. By using a network of multiple satellites, these satellites will provide real-time global observations, greatly increasing the amount of available data for NWP centers and are expected to further improve the accuracy of weather forecasts.

## Code and Data availability

COSMIC-2 and Metop-C RO data are available at www.cosmic.ucar.edu. YUNYAO data can be shared offline. Processed data and code can be downloaded from https://doi.org/10.5281/zenodo.13374107.

## Author contributions

XX, WH, and ZG contributed to the development of the ideas. XX, WH, and JW conducted data analysis. XX and WH wrote the paper. FL, NF, and YC provide data and revise paper.

## Competing interests

The contact author has declared that none of the authors has any competing interests.

## Disclaimer

Publisher's note: Copernicus Publications remains neutral with regard to jurisdictional claims made in the text, published maps, institutional affiliations, or any other geographical representation in this paper. While Copernicus Publications makes every effort to include appropriate place names, the final responsibility lies with the authors.

## Acknowledgements

We appreciate the China Meteorological Administration, ECMWF and NCEP for the data. We are also very grateful to the reviewers for their careful review and very valuable comments.

## Financial support

This research was funded by the National Natural Science Foundation of China (42175082, and 42075155) and the National Key Research and Development Program of China (2022YFC3004004).

## Review statement

This paper was edited by Peter Alexander and reviewed by Richard Anthes and one anonymous referee.

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
