# Peer review of "Quality Assessment of YUNYAO GNSS-RO Data in the Neutral Atmosphere"

_Atmospheric Measurement Techniques, 2024_

## Author Comment (AC1)

**Reviewer #1:**

1. This is a well-written paper and presents interesting results evaluating radio occultation (RO) observations from a constellation of eight small commercial satellites developed by Tianjin Yunyao Aerospace Technology. The eight Yunyao satellites produced about 12,000 RO occultations per day over the period May-June 2023. The quality of these observations are evaluated based on mean and standard deviations of Yunyao refractivities computed by comparing with radiosondes and two models, as well as the penetration rates (% of profiles reaching levels near the Earth's surface). This paper is a potentially valuable contribution and should be published, but there are several areas where improvements are needed.

Response: We appreciate the reviewer's positive evaluation of our work. We have actively adopted your suggestions for the manuscript and made modifications in accordance with your requests.

2. The paper says Yunyao plans to launch 90 satellites, which would provide an unprecedented number of RO profiles per day (probably greater than 150,000). The impact of this large number of RO profiles on NWP forecasts is expected to be very large, and hence this paper, which is the first to describe the quality of Yunyao RO observations, is important. The readers would be interested in the timeline for these launches and the likelihood that they will actually be launched. I know this is a hard question, but can the authors comment on this? Are the resources already obtained or are they merely a goal? Also, will all 90 satellites be launched into similar polar orbits, or will some be launched in low-inclination orbits? Some additional detail on the plans and their status would be useful.

Response: We are very grateful for your comments. We have further provided a timeline for the satellite launches (Table 2; Lines 51). The information in the table can serve as a reference for the launch schedule, but the actual launch dates may vary.

3. The methodology for evaluating the Yunyao observations in this paper is sound and the statistics look comparable to those from other RO missions. However, the paper only evaluates refractivities, and most current NWP models assimilate bending angles (BA). The paper would be improved if it evaluated BA rather than refractivities. The authors should explain why they chose refractivities instead of BA. If possible, they should include at least some BA evaluations in the paper.

Response: We agree with your comment. The original manuscript primarily focused on evaluating refractivity because CMA currently assimilates refractivity as the primary variable. Based on your suggestion, we further evaluated bending angles. Using the one-dimensional forward model in the Radio Occultation Processing Package (ROPP), we converted ERA5 variables into bending angles and compared them with the observations (Figures 5, S1). We have added a description of the methods in Sections 2.2.1 and 2.2.2, and included the results in Section 3.1 (lines 248-257).

4. The paper does not describe or provide a reference on how the Yunyao data are processed. Early studies of Yunyao BA data, which were graciously provided by Yunyao for the Radio Occultation Modeling Experiment (ROMEX), showed that there were issues in the BA uncertainties around 12 km, thought to be associated with the transition from geometric to wave optics. There was another issue of large standard deviations and large negative biases below 5 km in the BA, which was apparently due to very strict quality control in Yunyao's lower-level processing, where carrier phase measurements were removed from the excess phase data. There does seem to be large negative biases in the refractivities below 5 km in these results (Fig. 3); however, the refractivity error SD in these figures look reasonable. It would be good if the paper could summarize how the Yunyao data are processed and if the two issues above have been resolved in the current processing.

Response: We agree with your comment. Since the YUNYAO Data Processing Center introduced their GNSS-RO profile products, with the advice of Richard A. Anthes, Christian Marquardt and other experts, their GNSS-RO data processing methodology has been updated in three aspects. First, the deviation observed between 20 and 40 km, distinct from other GNSS-RO missions, was resolved by adjusting the smoothing window width for the exceed phase-to-Doppler inversion to optimize its adaptability to YUNYAO's high-sampling-rate data (100 Hz). Second, to address the sudden increase in SD below 12 km, YUNYAO investigated the open-/close-loop transition algorithms employed in other GNSS-RO missions. Their retrieval chain automatically identified the L1 open-/closed-loop splicing points through the L2 lock marks and used a sigmoid function as a weight to ensure a smooth transition from L1 closed-loop observations to open-loop observations. Third, for altitudes below 5 km, YUNYAO redesigned the L1 data truncation strategy to use the complete L1 open-loop observations as much as possible and process to obtain continuous exceed phases. In the geometric optics retrieval process, the Doppler retrieval truncation strategy is implemented by

identifying cases where the difference between the Doppler shift obtained from L1 and that from the empirical atmospheric model exceeds a specified threshold. In the wave optics retrieval process, the effective bending angle sequence is obtained by restoring the signal amplitude from the exceed phase at each height in the full spectrum inversion of the bending angle retrieval. The above strategies have dealt well with the preliminary product problems provided by YUNYAO in the ROMEX experiment. We also add descriptions in the manuscript (lines 68-80).

5. It would also be useful to include estimates of the error SD for the refractivity data up to 50-60 km, using the ERA5 and FNL model analyses as the other two datasets (extend Figs. 6-8 up to 50-60 km).

Response: Thank you very much for your suggestions. We apologize for not being able to provide results for the 50–60 km altitude range, as the ERA5 pressure-level data does not include atmospheric information at such high altitudes. The figure below shows the comparison between observations and ERA5 within the 0–50 km altitude range. As shown in the figure, the number of observations available for comparison decreases rapidly in the 40–50 km range, with almost no matching observations near 50 km, resulting in anomalous mean bias calculations. Therefore, we only presented results below 40 km in the manuscript. For Figs. 6–8 in the original manuscript, since we only obtained FNL data for the 26 mandatory levels (from 1000 hPa to 10 hPa), it was challenging to extend the results to higher atmospheric altitudes.

[Figure]

6. As the paper notes, the radiosonde 3CH uncertainty estimates are less than those of the RO, which is different from the results of Schreiner et al. 2020 and Rieckh et al. 2021. They attribute this difference to the fact that they used a double-differencing correction to the radiosonde data, which is plausible since Rieckh et al. and Schreiner et al. did not. They could test their proposed reason easily be redoing their Fig. 5 with uncorrected radiosonde data and present the results in a second part to Fig. 5.

Response: Thank you for the suggestion. We have added the results without using spatial-temporal sampling correction (Fig. 6) and revised the relevant descriptions (lines 264–277). We found that the spatial-temporal sampling correction is indeed the main reason for the differences between our results and those of Schreiner et al. (2020) and Rieckh et al. (2021). For the results above 10 km, we speculate that the differences may be caused by variations in quality control.

7. The length of the paper is appropriate, and the quality of the figures overall is high, with a few exceptions (Fig. 2 and Fig. 4). Please see detailed comments on these figures.

In summary, this paper could be acceptable for publication after the authors consider the above comments. I look forward to seeing a revised version.

Response: Thank you for the suggestion. We have made revisions to the relevant content based on your suggestions (responses provided below).

8. Lines 45-46—The statement that CMA-GFS incorporates about 20,000 RO profiles per day is interesting, and it would be useful to summarize the impact of these observations and refer to a study that shows this impact if one exists. By coincidence, this number is the number that the WMO International Working Group on Radio Occultation (IROWG) has been recommending for many years now for an operational RO backbone. Are there any published studies or reports that discuss the impact of these observations on the CMA forecasts?

Response: Thank you for the suggestion. We sincerely apologize that no study has systematically evaluated the impact of these observations. However, we have provided the source of the 20,000 profiles (Table 1).

9. Lines 69-70—I am sure the three references here do not demonstrate that the Yunyao receivers are "significantly smaller and lighter" than the COSMIC and Metop/GRAS receivers. I don't doubt that they are, but some numbers should be given, i.e.how much smaller and lighter?

Response: Thank you for the suggestion. We sincerely apologize that the references indeed do not contain information about the size and weight of the COSMIC and Metop/GRAS receivers. However, Sun et al. (2018) provides a description of the weight of the FengYun-3C/GNOS. We have revised the content (lines 70-72).

10. The penetration depths (minimum heights above ground reached by the RO profiles) shown in Fig. 2 are not very clear. For example, the color difference between 500 m and 2 km (a large difference in penetration depth) is very small, and it is difficult to distinguish between them. I suggest contours of penetration depth with a contour interval of 250 or 500 m for these figures, probably after the data in the grid boxes are smoothed. Alternatively, plot cumulative % profiles of penetration depths of zonal averages of the profiles for several representative latitudes for the GPS, GLONASS and BDS, similar to Fig. 2 of Schreiner et al. (2020).

Response: Thank you for the suggestion. We have redrawn Figure 2, modified the colorbar, and set

a color interval of 500 meters. We believe that the current figure more clearly displays the values in different regions. We did not apply smoothing or draw contour lines because the grid resolution of the figure is relatively low, and smoothing could distort the results.

11. Line 183—Give the value of the mean bias between 4-40 km, and also 10-30 km (the sweet spot for RO).

Response: Thank you for the suggestion. We have made modifications based on your suggestions (lines 220-223).

12. 4-Similar to the comment for Fig. 2, the colors for the middle panel (SD) do not show the variations clearly (everything is dark blue) Again, contours of the smoothed data would be clearer.

Response: Thank you for the suggestion. We have redrawn Figures 4 and 1 and modified the colorbar. We believe the current figures more clearly display the values at different locations.

13. 4, right panel: Why is the latitude distribution of numbers not symmetric? The are many more between 25-40°S than 25-40°N.

Response: The YUNYAO satellite's data transmission to the ground is primarily dependent on ground stations located within China. During the data transmission process, the satellite is required to execute specific onboard operations, thereby reducing the number of occultation observations over China and its surrounding areas, as well as throughout its entire trajectory from the United States into China. We have added the relevant content to the manuscript (lines 93-97).

14. Line 192: These are fairly small differences, and may not be important for DA. Also, most models assimilate BA not N.

Response: We have removed this part of the description (line 229).

15. Line 211: Fig. 4 should be Fig. 5.

Response: Thank you for your careful review. We have made modifications in the manuscript (line 265).

16. Line 227: "is" should be "are.

Response: Thank you for your careful review. We have made modifications in the manuscript (line 287).

17. Line 253: Shouldn't 8a be 7a?

Response: Thank you for your careful review. We have made modifications in the manuscript (line 313).

18. Line 314: Earlier you said the double differencing was the likely reason for the radiosondes SD being less than the RO SD, and I think this is probably the main reason. You should mention this here. Or, since this is not a major conclusion for this study, you could delete reference to it in the Summary and Conclusions section.

Response: Thank you for your careful review. We have added the relevant content to the Summary and Conclusions section (lines 374-376).

---

## Author Comment (AC2)

**Reviewer #2:**

1. This manuscript by Xu et al assess the mean difference, standard deviation and error for eight different Yunyao satellites by comparing them to radiosondes, COSMIC-2 and Metop B/C observations. The main measure is retrieved refractivity. The study is very detailed and the figures are in a good shape. This paper demonstrates nicely the quality of this new dataset. However, I miss the assessment of bending angles instead of refractivity as this measure is most often used for the assimilation in NWP models. I accept the publication after addressing this issue. Further, I have the following minor issues:

Response: We appreciate your positive evaluation of our work. We have actively adopted your suggestions for the manuscript and made modifications in accordance with your requests. We further evaluated the bending angles. We used the one-dimensional forward model in the Radio Occultation Processing Package (ROPP) to transform the ERA5 variables into bending angles and compared them with the observations (Figures 5, S1). We have added a description of the methods in Sections 2.2.1 and 2.2.2, and included the results in Section 3.1 (lines 248-257).

2. page 1, l.30: I wouldn't phrase that refractivity is a function of liquid and frozen water – of course if you have a polarized signal than the polarisation would be affected but not the bending of the ray path. Of course, super refraction can occur in stratocumulus regions.

Response: Thank you for the suggestion. We have removed "liquid water content and ice water content" (line 30).

3. page 2, l.46. It would be good to know what data goes already into CMA-GFS. 20.0000 daily profiles is impressive. Maybe a small table would help.

Response: Thank you for the suggestion. We have provided the source of the 20,000 profiles (Table 1).

4. page 2, l.47: Does GeoOptics still provides data in year 2024?

Response: We have verified that GeoOptics data does not appear to be included in CMA at present and have made modifications in the manuscript (lines 49).

5. page 6, l.126/127: This phrase summarizes very shortly how one derives e.g. refractivity or also physical measures, like temperature. It is good to mention that one has to make certain assumptions to get there. Probably good to add this here.

Response: Thank you for the suggestion. We have added the relevant description to the manuscript (lines 150-152).

6. page 7, l.142: How is the interpolation done in the vertical? Linearly or doing a spline interpolation?

Response: We used linear interpolation. A more detailed description is provided in Section 2.2.2 of the manuscript (lines 165-175).

7. page7, l.146/7: Here, I am slightly confused. Which method to you use to calculate MB and SD? The method by Lanzante or eq 2,3 and 4. I guess you use Lanzante to get rid off outliers and then use this cleaned sample to compute MB, SD according to the given equations. Probably rephrasing this sentence, makes this clearer.

Response: Thank you for the suggestion. We have made modifications in the manuscript (lines 177-180).

8. p.12, l.216: instead of *will be* write *is*

Response: Thank you for your careful review. We have made modifications in the manuscript (line 269).

---

## Referee Report (RR1)

5 January 2025
Second Review of Xu et al. 2024 (v3)

Richard Anthes

Xu, Xiaoze, Wei Han, Jincheng Wang, Zhiqiu Gao, Fenghui Li, Yan Cheng,and  Naifeng Fu, 2024: Quality Assessment of YUNYAO GNSS-RO Refractivity Data in the Neutral Atmosphere. *Atmos. Meas. Tech.*
*https://doi.org/10.5194/amt-2024-150*

This version is improved over the first version, which was already very good, and the authors have been responsive to the two reviews. The addition of the information on how the processing of the Yunyao data was improved since the original evaluation of the data provided to ROMEX about one year ago is a good addition. The paper is now acceptable for publication after the authors consider a few more comments. The authors may respond as they wish to each one, but they are not requirements before the paper is published.  I do not need to review the paper again.

1.  The authors have added some welcome details concerning the future launches of Yunyao RO satellites, and a few more details would be interesting if it is not too difficult. First, please give an estimate of the total number of RO profiles per day that are expected from the 90 satellites in lines 13, 56, and 352. Second, the sentence in line 354 "As of September 25, 2024..." should be updated with a more recent date in 2025. A timeline showing the number of RO profiles per day as new launches occur and previous satellites die would be interesting, but it is not necessary for the publication of this paper.

2.  The statement in lines 94-97 is interesting: *"Notably, the YUNYAO satellite's data transmission to the ground is primarily dependent on ground stations located within China. During the data transmission process, the satellite is required to execute specific onboard operations, thereby reducing the number of occultation observations over China and its surrounding areas, as well as throughout its entire trajectory from the United States into China."* However, I do not see a noticeable reduction in numbers of RO from GPS (b) and GLONASS (d) in Fig. 2. There are noticeable reductions (dark blue) over the middle east/eastern Europe and Indonesia in all three figures (b,d, and f), but I don't see any noticeable reduction over China and between China and the US in figures 2b and 2d.

3.  Section 2.1.3: The authors used a limited, low-resolution version of the ERA5 reanalysis, with 37 layers and top at 1 hPa (about 47 km). The full ERA5 dataset consists of 137 levels with a top at 0.01 hPa (80 km) https://cds.climate.copernicus.eu/datasets/reanalysis-era5-complete?tab=overview.  This means they cannot compare RO data with ERA5 above about 40 km and the low resolution produces a wavy structure in some of the

profiles (e.g. Fig. 5), as the author note. The authors should acknowledge that they are using a low-resolution version of ERA5 and give a reason why they don't use the full version.

4. In Fig. 8b, the Yunyao N biases between 20 and 40 km of ~-0.5% with respect to ERA5 look a bit too large compared to what we find in the ROMEX data. The biases in Fig. 8a are much closer to zero and are more like what we find. This is just a comment for the authors to consider. It looks like the sample in different in Fig. 8b compared to 8a. Are there other reasons?

In the Summary, the authors may want to revise the sentence to focus on the 8-35 km layer where the observations have much more effect on NWP models: "*Compared to the refractivity calculated from ERA5, the absolute value of the mean bias (MB) of YUNYAO RO refractivity and bending angle data within the 0–40 km range are less than 1.54% and 4.51%, respectively, with larger biases mainly occurring in the lower troposphere. The negative bias in the lower troposphere has been extensively discussed in previous studies (Sokolovskiy et al., 2014; Xie et al., 2010). The standard deviation (SD) of refractivity and bending angle data between 0 and 40 km are less than 3.35% and 11.06%, respectively, with larger values mainly found in the lower troposphere and upper stratosphere.*" The biases and SD are much smaller in this layer.

Minor editorial comments

Line 9: "is" should be "are"

Line 86: Caption to Table 3-I suggest replacing "main parameters" with "Characteristics".

Line 105-replace "at bottom atmosphere" with "in the lower troposphere"

Line 107-replace "is greater" with "is not as deep."

---

## Author Response (AR2)

Dr. Peter Alexander and Richard Anthes

We would like to thank you for your careful reading, valuable comments, and constructive suggestions, which have greatly improved the presentation of our manuscript. We have carefully considered Richard Anthes' comments and revised the manuscript accordingly. The manuscript has also been thoroughly checked, and any typos and grammatical errors found have been corrected. We believe that our responses have effectively addressed all the concerns raised by the reviewers. Not only that, we also revised the color schemes of Figures 3, 5, 6, 7, 8, and 9 to better ensure that the color schemes of the lines allow readers with color vision deficiencies to correctly interpret our findings. To better highlight the differences, Figures 1, 2, and 4 were not further modified. Although Figures 1, 2, and 4 are not friendly to individuals with monochromacy, the relevant descriptions in the main text help in understanding these figures.

1. This version is improved over the first version, which was already very good, and the authors have been responsive to the two reviews. The addition of the information on how the processing of the Yunyao data was improved since the original evaluation of the data provided to ROMEX about one year ago is a good addition. The paper is now acceptable for publication after the authors consider a few more comments. The authors may respond as they wish to each one, but they are not requirements before the paper is published. I do not need to review the paper again.

The authors have added some welcome details concerning the future launches of Yunyao RO satellites, and a few more details would be interesting if it is not too difficult. First, please give an estimate of the total number of RO profiles per day that are expected from the 90 satellites in lines 13, 56, and 352. Second, the sentence in line 354 "As of September 25, 2024..." should be updated with a more recent date in 2025. A timeline showing the number of RO profiles per day as new launches occur and previous satellites die would be interesting, but it is not necessary for the publication of this paper.

Response: We appreciate your comments. Due to potential changes in satellite orbits and signal reception, we are currently unable to accurately estimate the number of RO profiles that can be obtained daily from future satellite launches. In line 91, we have described the number of RO profiles that can currently be obtained from a single satellite, which helps to roughly estimate the number of RO profiles that can be obtained from 90 satellites. The sentence in line 354 has been

modified to "As of 2024, 30 satellites have been successfully launched" and Table 2 has been updated accordingly.

2. The statement in lines 94-97 is interesting: "Notably, the YUNYAO satellite's data transmission to the ground is primarily dependent on ground stations located within China. During the data transmission process, the satellite is required to execute specific onboard operations, thereby reducing the number of occultation observations over China and its surrounding areas, as well as throughout its entire trajectory from the United States into China." However, I do not see a noticeable reduction in numbers of RO from GPS (b) and GLONASS (d) in Fig. 2. There are noticeable reductions (dark blue) over the middle east/eastern Europe and Indonesia in all three figures (b, d, and f), but I don't see any noticeable reduction over China and between China and the US in figures 2b and 2d.

Response: We appreciate your comments. We have revised the description in this section (Lines 93-98).

3. Section2.1.3: The authors used a limited, low-resolution version of the ERA5 reanalysis, with 37 layers and top at 1 hPa (about 47 km). The full ERA5 dataset consists of 137 levels with a top at 0.01 hPa (80 km) https://cds.climate.copernicus.eu/datasets/reanalysis-era5-complete?tab=overview. This means they cannot compare RO data with ERA5 above about 40 km and the low resolution produces a wavy structure in some of the profiles (e.g. Fig. 5), as the author note. The authors should acknowledge that they are using a low-resolution version of ERA5 and give a reason why they don't use the full version.

Response: We appreciate your comments. We only used the 37-layer ERA5 reanalysis dataset and evaluated the data quality below 40 km. Since 40 km is above the layer (8-35 km) where observations exert a greater influence on NWP models, we consider the 37-layer ERA5 data to be sufficient. Numerous studies evaluating GNSS RO data primarily focus on data accuracy below 40 or 45 km (Cucurull et al., 2007; Ho et al., 2023; Sun et al., 2018). The wavy structure in Figure 5 is attributable to the lower vertical resolution of the ERA5 data. However, this phenomenon may also occur even when using the 137-layer dataset (Schreiner et al., 2020, Fig. 5). In the subsequent phase of our research, we will employ model-level data with higher vertical resolution to evaluate the

quality of GNSS-RO data at higher altitudes and examine its impact on the assimilation system.

4. In Fig.8b, the Yunyao N biases between 20 and 40 km of ~-0.5% with respect to ERA5 look a bit too large compared to what we find in the ROMEX data. The biases in Fig. 8a are much closer to zero and are more like what we find. This is just a comment for the authors to consider. It looks like the sample in di\erent in Fig. 8b compared to 8a. Are there other reasons?

Response: Thank you for your comments. We believe that the discrepancy in biases between Fig. 8a and Fig. 8b is primarily due to differences in the sample sets. Fig. 8a uses global observations, whereas Fig. 8b uses observations limited to the region between 45°S and 45°N. No additional processing was performed. The latitudinal differences in biases can be seen in Fig. 4.

5. In the Summary, the authors may want to revise the sentence to focus on the 8-35 km layer where the observations have much more effect on NWP models: "Compared to the refractivity calculated from ERA5, the absolute value of the mean bias (MB) of YUNYAO RO refractivity and bending angle data within the 0–40 km range are less than 1.54% and 4.51%, respectively, with larger biases mainly occurring in the lower troposphere. The negative bias in the lower troposphere has been extensively discussed in previous studies (Sokolovskiy et al., 2014; Xie et al., 2010). The standard deviation (SD) of refractivity and bending angle data between 0 and 40 km are less than 3.35% and 11.06%, respectively, with larger values mainly found in the lower troposphere and upper stratosphere." The biases and SD are much smaller in this layer.

Response: Thank you for your comments. We have added a description of the MB and SD between 10 and 30 km (Lines 357-362). The selection of 10–30 km is made to maintain consistency with the description in the main text.

6. Line 9: "is" should be "are"

Response: Thank you for your careful review. We have made modifications in the manuscript (Line 9).

7. Line 86: Caption to Table 3-I suggest replacing "main parameters" with "Characteristics". Line

Response: Thank you for your careful review. We have made modifications in the manuscript (Line

8. 105-replace "at bottom atmosphere" with "in the lower troposphere"

Response: Thank you for your careful review. We have made modifications in the manuscript (Line 106).

9. Line 107-replace "is greater" with "is not as deep."

Response: Thank you for your careful review. We have made modifications in the manuscript (Line 109).